

# Comprehensive review of dimensionality reduction algorithms: challenges, limitations, and innovative solutions

Aasim Ayaz Wani

School of Engineering, Cornell University, Ithaca, New York, United States

## ABSTRACT

Dimensionality reduction (DR) simplifies complex data from genomics, imaging, sensors, and language into interpretable forms that support visualization, clustering, and modeling. Yet widely used methods like principal component analysis, t-distributed stochastic neighbor embedding, uniform manifold approximation and projection, and autoencoders are often applied as "black boxes," neglecting interpretability, fairness, stability, and privacy. This review introduces a unified classification—linear, nonlinear, hybrid, and ensemble approaches—and assesses them against eight core challenges: dimensionality selection, overfitting, instability, noise sensitivity, bias, scalability, privacy risks, and ethical compliance. We outline solutions such as intrinsic dimensionality estimation, robust neighborhood graphs, fairness-aware embeddings, scalable algorithms, and automated tuning. Drawing on case studies from bioinformatics, vision, language, and Internet of Things analytics, we offer a practical roadmap for deploying dimensionality reduction methods that are scalable, interpretable, and ethically sound—advancing responsible artificial intelligence in high-stakes applications.

## INTRODUCTION

The proliferation of high-dimensional data across domains such as genomics, computer vision, NLP, finance, and environmental monitoring has made DR an essential component of modern data science workflows (*Meilă & Zhang, 2024*). By mapping high-dimensional datasets into lower-dimensional representations, dimensionality reduction (DR) techniques facilitate visualization, denoising, feature extraction, and pattern discovery. Moreover, they improve the performance and interpretability of downstream tasks, including clustering, classification, anomaly detection, and predictive modeling (*Ayesha, Hanif & Talib, 2020*).

Classical DR approaches, particularly linear techniques such as principal component analysis (PCA), linear discriminant analysis (LDA), and factor analysis (FA), offer efficiency, transparency by projecting data onto linearly defined subspaces (*Greenacre et al., 2022*; *Qu & Pei, 2024*). However, these models often fail to capture nonlinear

Corresponding author
Aasim Ayaz Wani,
aasimwani1@gmail.com

relationships and manifold structures that characterize real-world datasets. In response, nonlinear methods—such as t-distributed stochastic neighbor embedding (t-SNE), uniform manifold approximation, projection (UMAP), Isomap, and locally linear embedding (LLE) have emerged to better preserve local and global topology (*Healy & McInnes, 2024*). In parallel, deep learning-based DR methods, including autoencoders (AEs), variational AE (VAEs), transformer-based embeddings, have extended the field to support generative modeling and complex representation learning (*Kingma & Welling, 2019*; *Asperti & Trentin, 2020*).

Despite these advances, DR techniques are frequently deployed as black-box tools with minimal attention to key methodological concerns. Crucial questions—such as how many dimensions to retain, how to ensure embedding stability, how to interpret latent representations, and how to mitigate bias or privacy risks—often go unaddressed (*Kobak & Linderman, 2021*). This oversight poses serious challenges in high-stakes applications like precision medicine, financial forecasting, and legal analytics, where transparency, reproducibility, and ethical compliance are non-negotiable.

This review provides a comprehensive and critical synthesis of DR methods, organized into four main categories: linear, nonlinear, hybrid, and ensemble approaches. We identify and examine eight persistent challenges that constrain real-world applicability: dimensionality selection (DS), overfitting, instability, interpretability, scalability, bias propagation, noise sensitivity, and ethical risks such as reversibility and re-identification (*Greenacre et al., 2022*). For each, we outline state-of-the-art solutions, including intrinsic dimensionality estimation, fairness-aware and privacy-preserving embeddings, robust graph-based methods, and scalable deep architectures (*Marukatat, 2023*,; *Kingma & Welling, 2019*). By linking theory with practical use cases and implementation considerations, this review aims to equip researchers and practitioners with the tools and strategies needed to apply DR techniques effectively.

# CLASSIFICATION OF DIMENSIONALITY REDUCTION METHODS

Selecting an appropriate dimensionality–reduction (DR) technique is pivotal for revealing meaningful structure in high-dimensional data. Each method encodes assumptions about the data geometry—linearity, neighborhood continuity, smooth manifolds—that must align with the downstream objective. Formally, DR maps a matrix $\mathbf{X} \in \mathbb{R}^{n \times d}$ to an embedding $\mathbf{Y} \in \mathbb{R}^{n \times k}$ with $k \ll d$, while striving to preserve global variance, local topology, or class separability. As illustrated in Fig. 1, different DR algorithms emphasize distinct structural properties, resulting in varied geometric interpretations of the same manifold.

## Linear approaches

Linear techniques project data onto low-dimensional subspaces. PCA identifies orthogonal directions of maximal variance, offering speed and interpretability (*Greenacre et al., 2022*; *Jolliffe & Cadima, 2016*). PCA underpins exploratory plots, gene-expression compression, and sensor decorrelation, but struggles with strong non-linearities or outliers (*Bian et al.,*

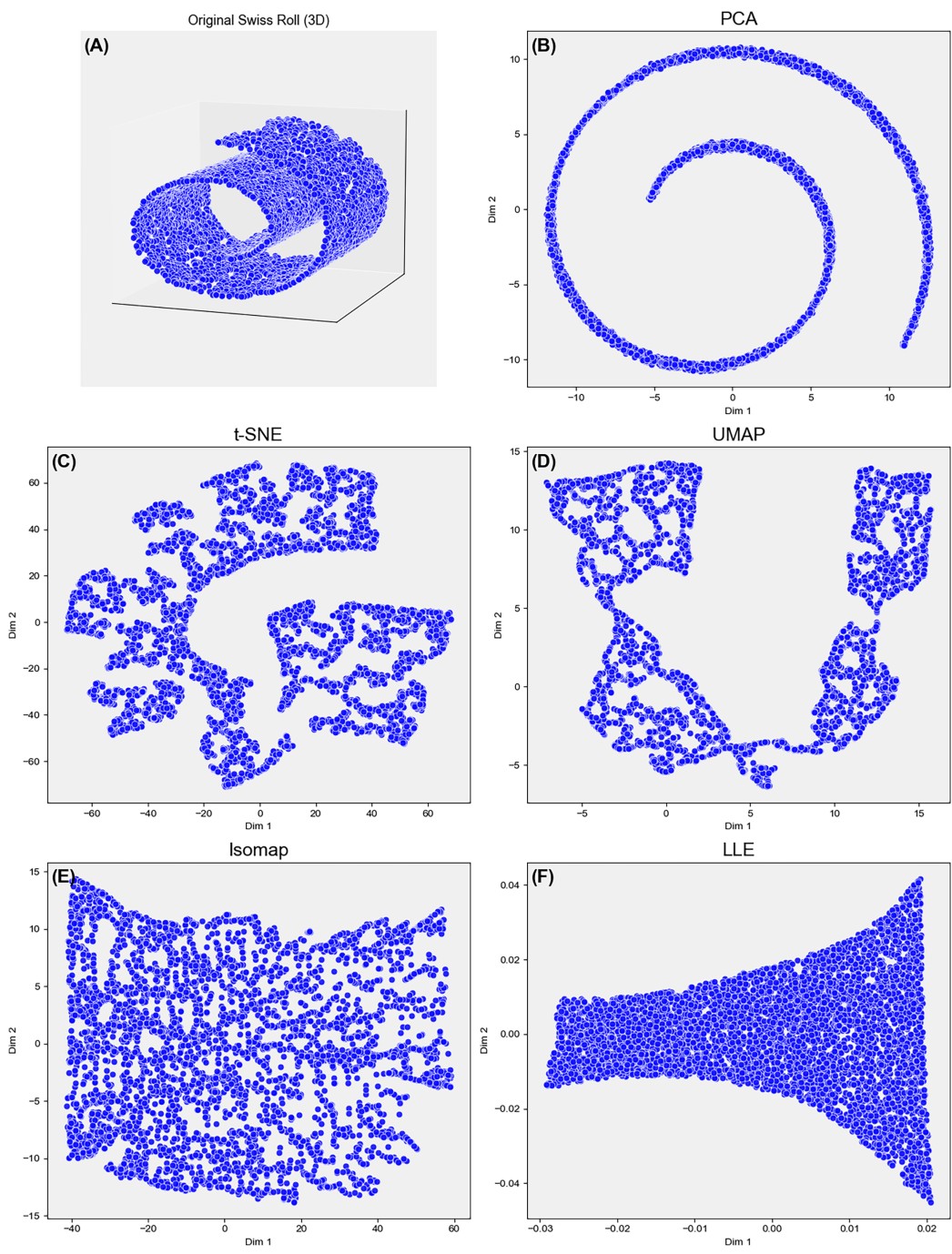

**Figure 1 Comparison of five DR methods–on the 3D Swiss Roll projected to 2D.** (A–F) PCA fails to unfold the nonlinear manifold. UMAP and Isomap preserve both local and global structures, with UMAP yielding a cleaner unroll. t-SNE captures local neighborhoods but distorts global structure. LLE poorly recovers global geometry, highlighting trade-offs in structure preservation across methods.

*2022*). LDA optimises between-/within-class scatter for supervised tasks such as face recognition or biomarker discovery, yet assumes class homoscedasticity and balanced priors (*Li et al., 2023a*; *Qu & Pei, 2024*; *Li et al., 2024*). FA decomposes observed variables

**Table 1  Summary of linear dimensionality reduction variants.**

| Variant | Description | References |
|---|---|---|
| *Principal component analysis (PCA) variants* | | |
| Standard PCA ($O(nd^2)$) | Projects data along directions of maximum variance. Fast and interpretable baseline, but fails on nonlinear manifolds and is sensitive to feature scaling. | *Abdi & Williams (2010)*, *Jolliffe & Cadima (2016)* |
| Sparse PCA ($O(ndk)$) | Adds $\ell_1$ penalty to promote sparse loadings, improving interpretability. Requires careful tuning and may reduce numerical stability. | *Li et al. (2023b)*, *Xiao et al. (2023)* |
| Robust PCA ($O(nd \log d)$ or higher) | Decomposes input into low-rank and sparse components, making it robust to noise and outliers. Computationally expensive on large datasets. | *Gao et al. (2021)*, *Bian et al. (2022)* |
| Kernel PCA ($O(n^3)$) | Uses kernel trick to capture nonlinear structure in high-dimensional feature space. Models curvature well, but interpretability is poor and performance depends on kernel choice. | *Shahzad, Huang & Memon (2022)*, *Fang et al. (2025)* |
| Probabilistic PCA ($O(nd^2)$) | Applies Bayesian PCA with Gaussian noise modeling. Enables uncertainty quantification but is sensitive to assumptions. | *Hong et al. (2021)*, *Collas et al. (2021)* |
| Incremental PCA ($O(ndk)$) | Performs PCA in mini-batches, improving memory efficiency for large or streaming data. May reduce accuracy and is sensitive to data order. | *Balsubramani, Dasgupta & Freund (2013)*, *Ross et al. (2008)* |
| Multilinear PCA ($O(n \prod_{m=1}^{M} d_m)$) | Extends PCA to tensor data *via* mode-wise decomposition. Captures multi-modal structure but has high computational cost and is sensitive to tensor shape. | *Guo, Zhou & Zhang (2021)*, *Han et al. (2023)* |
| *Linear discriminant analysis (LDA) variants* | | |
| Standard LDA ($O(nd^2 + d^3)$) | Maximizes class separation based on between- and within-class scatter. Effective for supervised DR but assumes equal class covariances and linear decision boundaries. | *Fisher (1936)*, *Hastie et al. (2009)* |
| Regularized LDA ($O(nd^2)$) | Stabilizes LDA with shrinkage for small-sample or noisy settings. Improves generalization but depends heavily on regularization parameter $\lambda$. | *Li et al. (2024)*, *Guo, Hastie & Tibshirani (2007)*, *Zaib et al. (2021)* |
| Kernel LDA ($O(n^3)$) | Projects data into kernel space before LDA to model nonlinear separation. Handles curved class boundaries but is hard to interpret and sensitive to kernel choice. | *Fang et al. (2025)*, *Li et al. (2023a)*, *Qu & Pei (2024)* |
| Sparse LDA ($O(ndk)$) | Adds sparsity to projection vectors for feature selection in high-dimensional spaces. Enhances interpretability but may be unstable under multicollinearity. | *Li et al. (2023c)*, *Park, Ahn & Jeon (2022)* |
| Penalized LDA ($O(nd^2)$) | Applies $\ell_2$ regularization to reduce overfitting. Useful in noisy or sparse data but may oversmooth projections. | *Kwon et al. (2024)*, *Wu, Wu & Wu (2021)* |
| Hierarchical LDA ($O(nd + T)$) | Models class hierarchies using tree-structured priors. Supports multi-level classification but requires complex training. | *Yu et al. (2019)*, *Wallach, Mimno & McCallum (2009)* |
| *Locally linear embedding (LLE) variants* | | |
| Standard LLE ($O(n^2d + nk^3)$) | Reconstructs each point as a linear combination of neighbors to preserve local geometry. Effective for unfolding manifolds but highly sensitive to noise and sampling. | *Chen & Liu (2011)*, *Xue, Zhang & Qiang (2023)* |
| Modified LLE ($O(n^2d + nk^3)$) | Adjusts weights for uneven sampling. Improves robustness under irregular density but lacks guarantees on global structure. | *Zhang & Wang (2006)* |
| Hessian LLE ($O(nk^4)$) | Incorporates curvature *via* Hessian penalty to capture second-order geometry. Effective for image manifolds but computationally intensive and fragile. | *Liu et al. (2022)* |
| Sparse LLE ($O(nk^2)$) | Adds $\ell_1$ constraints on reconstruction weights to highlight dominant neighbors. Improves interpretability but can become unstable in high correlation settings. | *Ziegelmeier, Kirby & Peterson (2017* |
| Geodesic LLE ($O(n^2 \log n)$) | Uses shortest path distances on neighbor graph to capture non-local structure. Suitable for complex topologies but sensitive to connectivity. | *Islam & Xing (2021)* |

into latent factors plus noise—valuable in psychometrics—but is restricted to linear signal models. Independent component analysis (ICA) and non-negative matrix factorisation (NMF) extend the linear family by enforcing statistical independence or non-negativity,

respectively, and excel in topic modelling (*Maćkiewicz & Ratajczak, 1993*; *Izenman, 2013*). (See Table 1 for Linear methods variant-wise breakdown.)

Yet real-world manifolds are rarely perfectly linear, motivating the nonlinear methods discussed next. Nonlinear approaches: These techniques uncover curved manifolds or high-order relations that linear projections overlook. They divide into manifold-learning algorithms and neural-network models. Manifold learning: t-SNE preserves local similarities and is standard for single-cell RNA-seq or word-embedding visualisation, but its perplexity sensitivity, global-structure distortion require caution (*Taylor & Merényi, 2022*; *Serna-Serna et al., 2023*). UMAP leverages fuzzy topological graphs to balance local and global faithfulness with lower runtime (*Healy & McInnes, 2024*; *McInnes, Healy & Melville, 2018*). Isomap retains geodesic distances; LLE maintains linear reconstructions in neighbourhoods—both effective for motion-tracking data yet noise-prone (*Xue, Zhang & Qiang, 2023*; *Chen & Liu, 2011*) (see Fig. 1). Kernel PCA lifts data into a reproducing-kernel Hilbert space before applying PCA, capturing nonlinear variance but incurring $O(n^2)$ memory for the kernel matrix; Nyström or random-feature approximations reduce this to $O(nm)$ ($m \ll n$) at modest accuracy cost (*Shahzad, Huang & Memon, 2022*). (See Table 2 for non-linear methods variant-wise breakdown.)

## Neural-network models

AEs learn encoder–decoder pairs that compress and reconstruct inputs, whereas VAEs add a probabilistic latent prior to enable generative sampling (*Asperti & Trentin, 2020*; *Kingma & Welling, 2019*). These models support image compression, multimodal fusion, and anomaly detection but demand large datasets, careful regularisation, and sacrifice transparency. Recent transformer-based encoders and self-supervised contrastive learners supply context-rich embeddings for vision and language. Fairness-regularised autoencoders further attempt to decorrelate sensitive attributes from latent codes (*Kingma & Welling, 2019*).

When neither linear nor a single nonlinear method suffices, hybrid and ensemble strategies provide a pragmatic compromise. Hybrid and Ensemble Approaches: hybrid pipeline applies PCA first–to denoise and decorrelate—followed by UMAP or t-SNE for nonlinear refinement, boosting scalability and stabilising initialisation on large image or single-cell datasets (*Kobak & Linderman, 2021*). Ensemble DR aggregates multiple embeddings from different seeds, subsets, or algorithms. Procrustes alignment, geometric averaging, or consensus fusion produce a robust embedding that mitigates run-to-run variance. The trade-off is higher computation and reduced interpretability of the consensus.

## Outlook

DR has progressed from linear decompositions to GPU-accelerated, self-supervised architectures. Linear methods endure for their speed and clarity; nonlinear and neural techniques reveal fine-grained patterns. Hybrid pipelines and ensembles dominate production workflows, balancing accuracy, stability, and transparency. Future advances

**Table 2 Summary of non-linear dimensionality reduction variants with time complexity.**

| Variant (time complexity) | Description: key idea, strengths, and limitations | References |
|---|---|---|
| **t-SNE variants** | | |
| Standard t-SNE ($O(N^2)$) | Preserves local neighborhood structure using probabilistic similarity in a low-dimensional space. Widely adopted for visualizing complex datasets, but it is non-invertible, distorts global relationships, and scales poorly. | *Van der Maaten & Hinton (2008), Serna-Serna et al. (2023)* |
| Barnes–Hut t-SNE ($O(N \log N)$) | Accelerates standard t-SNE *via* tree-based approximations, reducing computational complexity. Scalable for large datasets like single-cell RNA-seq, but approximation may distort dense regions. | *Van Der Maaten (2014), Meyer, Pozo & Zola (2021)* |
| Parametric t-SNE ($O(N)$ at inference) | Learns an explicit neural mapping from inputs to low-dimensional space. Enables reuse across datasets and supports transfer learning but sacrifices interpretability and requires retraining on distribution shift. | *Sainburg, McInnes & Gentner (2021), Chen et al. (2024)* |
| Joint t-SNE ($O(MN^2)$, M = modalities) | Embeds multiple modalities by enforcing a shared similarity structure across them. Useful in multi-view learning but assumes consistent structure and is vulnerable to modality-specific noise. | *Wang et al. (2021a), Taylor & Merényi (2022)* |
| Time-dependent t-SNE ($O(TN^2)$, T = timesteps) | Captures temporal structure by conditioning current embedding on previous state. Effective for visualizing trajectories or dynamic systems, though lacks theoretical stability guarantees. | *Ali, Borgo & Jones (2021), Linderman & Steinerberger (2022), Wang et al. (2021a)* |
| Accelerated t-SNE ($O(N \log N)$) | Employs GPU acceleration and algorithmic approximations for real-time or large-scale visualization. Performance is hardware-dependent and tuning-sensitive across implementations. | *Delchevalerie et al. (2021), Kang et al. (2021)* |
| **UMAP variants** | | |
| Standard UMAP ($O(N \log N)$) | Preserves both local and global structure using topological graph construction. Fast, unsupervised, and widely adopted, though it is non-invertible and sensitive to initialization. | *McInnes, Healy & Melville (2018), Ghojogh et al. (2023)* |
| Supervised UMAP ($O(N \log N)$) | Incorporates class labels to improve embedding coherence in supervised settings. Enhances class separation but may overfit on noisy or imbalanced labels. | *Becht et al. (2019), Kobak & Linderman (2021)* |
| Parametric UMAP ($O(N)$ at inference) | Trains a neural network to approximate UMAP embedding function, enabling transfer learning and embedding of new data. Reduces interpretability and requires retraining on distributional shifts. | *Sainburg, McInnes & Gentner (2021)* |
| Cross-entropy UMAP ($O(N^2)$) | Optimizes a divergence objective to improve alignment of high- and low-dimensional graphs. Preserves global structure but is slower and less stable on sparse graphs. | *Kobak & Berens (2019), Kobak & Linderman (2021)* |
| Density-preserving UMAP ($O(N \log N)$) | Adjusts embeddings to preserve input space densities, improving balance across populations. Effective in genomics and imbalanced datasets, but difficult to tune under sparse sampling. | *Narayan, Berger & Cho (2021)* |
| Metric UMAP ($O(N \log N)$) | Allows user-defined distance functions (*e.g.*, cosine, Jaccard) for more flexible embedding. Works well in NLP and recommender systems but degrades when metric mismatched to structure. | *McInnes, Healy & Melville (2018), Ghojogh et al. (2023)* |
| Temporal UMAP ($O(TN \log N)$) | Extends UMAP to encode sequence ordering, preserving continuity over time. Used in time-series and biological signal data but lacks a standard loss for temporal invariance. | *Sainburg, McInnes & Gentner (2021)* |

will stem from *AutoML-guided method selection*, *streaming manifold learners*, and *domain-aware adaptations* (*Xiao et al., 2023*). Concurrently, rising demands for interpretability, fairness, and privacy are steering research toward explainable, bias-controlled, and differentially-private DR algorithms. Rigorous benchmarks remain

essential for the responsible deployment of dimensionality-reduction techniques in high-impact settings (*Wang et al., 2021b*).

# CHALLENGES

## Determining the optimal dimensionality

Selecting the optimal number of dimensions ($k$) in a pipeline remains a persistent and unresolved challenge—one with broad implications for model performance, interpretability, and computational efficiency. With DR increasingly integral to high-stakes applications such as biomedical analytics, precision medicine, inadequate or arbitrary selection of dimensionality poses significant risks to the reproducibility, transparency, and reliability of downstream analyses (*Guerra-Urzola et al., 2021*).

### Under-reduction vs. over-reduction

The consequences of inappropriate DS manifest primarily as under-reduction or over-reduction. Under-reduction, where too few dimensions are retained, can obscure subtle but critical signals. For example, in single-cell RNA sequencing involving over 20,000 genes, reducing data to $k = 2$ for visualization may preserve less than 0.1% of total variance. This aggressive reduction can yield visually appealing clusters while collapsing biologically relevant gradients, such as differentiation trajectories or batch effects (*Yang et al., 2021*). Conversely, over-reduction—retaining dimensions beyond meaningful thresholds—can preserve irrelevant variance or amplify noise, especially in sparse, high-collinearity datasets found in NLP or cybersecurity. This typically inflates distance metrics and spurious correlations, leading to overfitting (*Narayan, Berger & Cho, 2021*).

### Heuristic-based selection and its pitfalls

Many dimensionality-selection practices rely on heuristics, such as retaining principal components that explain 90–95% of variance or using scree plots. Although computationally convenient, these methods assume variance equates to structure—an assumption often invalid in sparse or noisy data (*Greenacre et al., 2022*; *Jolliffe & Cadima, 2016*). Moreover, these heuristics are fundamentally linear and fail to translate to nonlinear DR techniques like t-SNE or UMAP, which lack variance-based metrics (*Kobak & Linderman, 2021*).

### Limitations of statistical thresholds

Statistical approaches like parallel analysis and Gavish–Donoho thresholding attempt to formalize DS using null distributions. While theoretically grounded, these methods often produce overly conservative estimates in real-world data, especially in noisy or limited-sample regimes. In scRNA-seq or text-mining, they may exclude biologically meaningful low-variance features (*Narayan, Berger & Cho, 2021*). Furthermore, their linear assumptions break down for deep or kernel-based DR models, where dimensions are learned *via* optimization rather than derived analytically.

### Challenges with intrinsic dimensionality estimation

ID estimators infer the minimal dimensions required to capture a dataset's manifold structure. Common ID methods include maximum likelihood estimation, correlation

dimension, and modern algorithms like DANCo and TwoNN (*Antwarg et al., 2021*). However, ID estimation is highly sensitive to noise, local density variations, and clustering artifacts. Estimates may vary substantially across similar subsets, and robust ID inference often requires large sample sizes and computationally expensive corrections (*Meilă & Zhang, 2024*).

### Task-dependence and dimensionality drift

Optimal dimensionality is task-specific. While 2D/3D embeddings suffice for visualization, high-dimensional embeddings are essential for classification, anomaly detection, or semantic search (*Marukatat, 2023*). Notably, performance often improves with increasing $k$ up to a point, then plateaus or deteriorates as noise dominates—a non-monotonic behavior. Moreover, optimal dimensionality can shift with dataset drift or evolving dimensionality drift, necessitating adaptive DR strategies such as continual learning and streaming methods (*Xiao et al., 2023*).

### Practical constraints on exhaustive dimensionality tuning

Exhaustively tuning $k$ *via* hyperparameter search is infeasible for many modern DR algorithms, especially nonlinear ones like t-SNE and UMAP, which are sensitive to initialization and stochasticity (*Wang et al., 2021b*). Advanced methods increasingly turn to AutoML and Bayesian optimization to infer $k$ implicitly. However, many neural approaches—*e.g.*, VAEs, SimCLR, and BYOL—control latent dimensionality indirectly *via* architecture rather than explicit hyperparameters, further complicating dimensionality tuning (*Kingma & Welling, 2019*).

### Summary

Choosing the right number of output dimensions k is crucial for balancing information retention and interpretability. Common methods include scree plots and explained variance for linear DR, with more robust options like parallel analysis and Gavish–Donoho thresholding for noise control. In high-dimensional data, intrinsic dimensionality estimators provide geometric insights but depend on sample density. Newer task-aware methods use contrastive loss to align k with downstream tasks. While each approach has trade-offs, together they help ensure embeddings preserve structure and usefulness.

## Navigating the interpretability–accuracy trade-off in dimensionality reduction

DR compresses high-dimensional data into compact forms that retain key features for tasks like clustering, classification *etc*. However, modern DR methods—especially those using nonlinear manifold learning—often trade interpretability for representational fidelity. This trade-off is especially problematic in sensitive domains where transparency, explainability, and regulatory compliance are essential.

### Defining the trade-off

We define *interpretability* in DR as the degree to which latent embedding structures can be explicitly mapped to original input features, domain-specific concepts, or human-understandable constructs. Conversely, *fidelity* pertains to preserving meaningful

relationships inherent to the data, such as global or local distance structures, neighborhood consistency, and class separability, essential for reliable downstream inference. Unfortunately, DR techniques offering high embedding fidelity typically employ complex nonlinear transformations that severely limit interpretability, while highly interpretable methods frequently underfit complex, nonlinear datasets (*Greenacre et al., 2022*).

### Opacity of nonlinear embeddings

Nonlinear manifold learning methods like t-SNE and UMAP are popular in bioinformatics, single-cell analysis, and computer vision for capturing complex topologies. However, their multistep embedding pipelines—neighbor graph construction, fuzzy set computation, and iterative optimization—obscure clear feature-to-embedding mappings. For instance, while UMAP effectively clusters cell types in single-cell RNA-seq, it can mask continuous biological gradients like differentiation (*Kobak & Linderman, 2021*; *Healy & McInnes, 2024*; *Narayan, Berger & Cho, 2021*). These methods are also sensitive to initialization and hyperparameters, raising reproducibility concerns.

### Deep latent representations and black-box risks

Deep learning–based DR methods—autoencoders, VAEs, transformer encoders—worsen interpretability by embedding data into nonlinear latent spaces with entangled feature dimensions (*Kingma & Welling, 2019*). In healthcare, this can blend sensitive traits (*e.g.*, race, income) with clinical variables, risking biased outcomes. Such opacity challenges fairness, transparency, and compliance with standards like General Data Protection Regulation (GDPR), HIPAA guidelines (*Zhang, Chen & Hong, 2021*).

### Transparency in linear models—but at what cost?

Linear DR methods preserve interpretability by projecting data into directions that are explicit linear combinations of the input features. These loadings allow direct mapping of latent dimensions to feature contributions, which is particularly useful in fields requiring justification, such as public health or forensic auditing (*Greenacre et al., 2022*). However, this transparency limits their capacity to model complex nonlinearity or curvature in the data manifold, rendering them ineffective in many real-world tasks involving temporal, sensory, or multimodal data.

### Regulatory and ethical implications

The interpretability–accuracy trade-off has significant ethical and legal implications. Global regulations—including the EU's GDPR, the proposed AI Act, and HIPAA—now demand accountability and traceability in automated systems. Failures in explainability, as seen in high-profile tools like COMPAS, have sparked scrutiny over DR's role in opaque modeling pipelines (*Mehrabi et al., 2021*). Embeddings that obscure feature contributions may undermine users' rights to explanation and compromise trust and compliance.

### Partial solutions and remaining gaps

While various methods have attempted to address this trade-off—such as hybrid pipelines, *post-hoc* attribution tools, and attention-based architectures—few offer generalizable, scalable, or domain-agnostic interpretability. Some strategies introduce significant

complexity or reduce performance, impeding adoption in real-world systems. For example, interpretability-enhancing models often lack robustness across noise-prone or sparse datasets (*Zhang, Chen & Hong, 2021*).

### Summary

Balancing interpretability and accuracy remains a key challenge in DR. While nonlinear models offer higher fidelity, they require new strategies to maintain interpretability. Future work should focus on unifying both goals through interpretable model designs, metrics for explanation quality, and standardized benchmarks for joint evaluation.

## Stability, overfitting, and generalization—the reliability triad

DR methods are increasingly embedded in scientific and industrial pipelines for visualization, clustering, anomaly detection, and representation learning. Yet, many of these methods—particularly nonlinear, deep, or graph-based approaches—are susceptible to a triad of interrelated problems: *stability*, *overfitting*, and *poor generalization*. Collectively, these issues undermine the reliability of embeddings, leading to inconsistent downstream decisions, analytic irreproducibility, and erosion of trust in model outputs (*Sainburg, McInnes & Gentner, 2021*; *Chen et al., 2024*).

### Instability and reproducibility failures

DR methods like t-SNE and UMAP involve stochastic optimization, non-convex losses, and approximate nearest neighbor (ANN) graphs, all of which introduce non-determinism (*Wang et al., 2021b*). As a result, repeated runs—even with the same data and parameters—can produce markedly different local and global structures. In single-cell RNA-seq, for instance, rerunning UMAP may merge or split cell clusters differently, affecting biological interpretation (*Kobak & Linderman, 2021*). Hyperparameter sensitivity worsens this: minor tweaks to perplexity (t-SNE) or neighbor count (UMAP) can reshape the manifold (*Taylor & Merényi, 2022*). Such variability undermines replicability, especially in exploratory analyses without ground truth.

### Overfitting in high-capacity DR models

Modern DR methods—especially autoencoders and VAEs—often use overparameterized architectures capable of memorizing training data. In high-dimensional, low-sample-size (HDLSS) settings, they risk capturing dataset-specific noise over meaningful structure (*Asperti & Trentin, 2020*; *Kingma & Welling, 2019*). For example, a deep autoencoder trained on limited EHR data may encode hospital-specific artifacts rather than general disease patterns. Despite low reconstruction loss, the latent space becomes noise-entangled, undermining downstream tasks. Even classical methods like PCA are vulnerable—under sparse, noisy conditions, leading components may align with outliers, distorting the true data trends (*Guerra-Urzola et al., 2021*; *Jolliffe & Cadima, 2016*).

### Generalization failures and domain fragility

Embeddings learned from one dataset often fail to generalize to distributionally shifted data due to covariate shift, batch effects, or sampling bias. In clinical contexts, PCA or autoencoder embeddings trained on one hospital's data may break down when applied to

another with different demographics or equipment. Even UMAP's transform function can distort class boundaries if new data deviate from the original density (*Zhang, Chen & Hong, 2021*). This domain fragility poses significant risks for real-time or cross-site deployment in diagnostics, fraud detection, or IoT streams.

### The rashomon effect in unsupervised embeddings

Unsupervised DR exacerbates the so-called Rashomon effect—where multiple, equally plausible embeddings reflect different underlying data aspects. Depending on initial conditions, loss weighting, or random sampling, models may emphasize demographic subgroups, measurement artifacts, or secondary trends. For example, an embedding of student performance data may cluster students by socioeconomic status in one run and by geographic region in another, despite both being technically valid. This ambiguity challenges interpretability and raises concerns about cherry-picking results that support preconceived narratives.

### Difficulty in evaluation and reporting

Despite the critical nature of these issues, robust evaluation of DR reliability is rarely performed. Unlike supervised models, DR lacks standardized generalization error metrics. Surrogate measures—such as Procrustes alignment, neighborhood preservation scores, or silhouette consistency across seeds—exist but are inconsistently applied and often omitted in publications. Without community-adopted benchmarks, method selection is *ad hoc*, and downstream conclusions may be built on unstable or misleading embeddings (*Mehrabi et al., 2021*).

### Summary

The reliability triad—instability, overfitting, and poor generalization—represents a core limitation of current DR pipelines. Each element reinforces the others: overfitting fuels instability; instability conceals overfitting; both degrade generalization. Addressing this requires holistic solutions—deterministic architectures, regularization, domain-aware constraints, and standardized evaluation—discussed in the next section as part of a roadmap toward robust, reproducible DR.

## Ethical concerns and reversibility risks

Although DR is often assumed to aid in anonymization by compressing high-dimensional datasets into abstract, low-dimensional forms, this assumption has increasingly come under scrutiny. Many DR techniques preserve enough structural detail to permit the re-identification of individual records or the inference of sensitive attributes (*Antwarg et al., 2021*). These vulnerabilities raise profound concerns when DR is applied in biomedical, financial, or behavioral domains, where confidentiality breaches can have legal, ethical, and societal consequences (*Mehrabi et al., 2021*).

### Inversion and reconstruction risks

Certain DR methods, such as PCA and autoencoders, are intrinsically reversible under known conditions. PCA applies orthogonal linear transformations, allowing approximate reconstruction when the projection matrix and retained components are available.

Autoencoders—including undercomplete and variational variants—explicitly train decoder networks to invert embeddings, often achieving high-fidelity recovery of input data (*Asperti & Trentin, 2020*; *Kingma & Welling, 2019*). In sensitive settings, this reversibility enables adversaries to reconstruct identifiable information from embeddings, thereby breaching data confidentiality (*Guerra-Urzola et al., 2021*).

### Attribute and membership inference from embeddings

Even in the absence of perfect reconstruction, low-dimensional embeddings can leak latent information. In collaborative ML workflows, such as federated learning or cross-institutional analytics, parties often exchange DR-transformed data under the assumption that abstraction ensures privacy. However, adversaries can leverage auxiliary knowledge to infer protected attributes, perform linkage attacks in training data—exposing a core weakness in the notion that DR inherently anonymizes data (*Zhang, Chen & Hong, 2021*). Such leakage is particularly troubling when demographic or health-related attributes are indirectly encoded in the embedding geometry.

### Lack of formal privacy guarantees

Mainstream DR algorithms—t-SNE, UMAP, PCA, autoencoders—lack formal privacy guarantees like differential privacy, $k$-anonymity, or $\ell$-diversity. Their focus on preserving data structure or neighborhood topology often conflicts with obfuscation goals. For example, UMAP's emphasis on class separability aids clustering but heightens vulnerability to attribute inference attacks (*Kobak & Linderman, 2021*). Without integrated privacy safeguards, such embeddings are prone to reverse engineering, especially in adversarial settings or with auxiliary data.

### Regulatory implications

The reversibility and latent leakage properties of DR methods create friction with data protection frameworks such as the GDPR, HIPAA. These regulations prohibit any transformations that allow re-identification of anonymized subjects or violate consent and data minimization principles. Yet most DR pipelines are implemented without compliance audits, interpretability guarantees, or transparency mechanisms, leaving organizations vulnerable to legal risk and public mistrust.

### Summary

The notion that DR inherently enhances privacy is common and flawed. Inversion attacks, attribute inference and the lack of formal privacy guarantees expose serious weaknesses in current DR practices. As DR methods are increasingly embedded in pipelines for healthcare, finance and behavioral analytics, addressing these ethical vulnerabilities becomes critical. Without robust mitigation strategies & regulatory accountability, DR may exacerbate privacy risks rather than mitigate them.

## Bias propagation from high-dimensional features

DR is frequently employed as a preprocessing step in analytical pipelines, often under the assumption that abstraction mitigates reliance on sensitive or confounding features. However, recent research demonstrates that DR techniques can retain—and even

amplify—structural biases embedded in the high-dimensional feature space. This is particularly problematic in domains where input features are often entangled with sensitive attributes such as race, gender, or socioeconomic status.

### Propagation of feature-level biases

DR methods aim to preserve intrinsic patterns within data but inevitably also retain statistical associations, including those involving sensitive variables. In applications like credit scoring or medical diagnostics, proxy features often correlate with protected attributes due to historical or systemic inequities. Even when explicit identifiers are excluded, nonlinear DR methods—such as t-SNE and UMAP—can produce latent spaces where demographic groupings remain separable, embedding socio-structural bias into ostensibly neutral representations (*Kobak & Linderman, 2021*).

### Structural biases in manifold construction

Manifold-based DR methods rely on constructing nearest-neighbor graphs to capture local geometry in high-dimensional space. If the data reflect underrepresentation, overdiagnosis, or sampling imbalance—as often occurs in clinical, legal, or educational datasets—those disparities become encoded in the graph structure itself. This topology influences the low-dimensional embedding, often reinforcing marginalization or over-clustering of minority subgroups (*Yang et al., 2021*; *Mehrabi et al., 2021*). In healthcare, for instance, diagnostic categories may be overrepresented for certain populations, resulting in distorted embeddings that mischaracterize patient.

### Downstream consequences of embedding bias

Biases introduced at the DR stage can propagate through the entire modeling pipeline. Downstream tasks such as clustering, stratification, recommendation, or prediction operate on biased representations, potentially leading to disparate outcomes. Because the embedding process is typically unsupervised and opaque, these biases may go undetected until decisions are already impacted. Even fairness-aware models applied later in the pipeline may be ineffective if the underlying representation space encodes distorted or discriminatory structure (*Ziegelmeier, Kirby & Peterson, 2017*).

### Summary

Bias propagation in DR is a critical yet often overlooked risk in modern data workflows. As DR is increasingly used in high-stakes domains, it can encode systemic disparities into opaque representations that influence key decisions. Unlike supervised models, DR methods are rarely audited for fairness. Addressing this gap requires bias-aware algorithms, fairness metrics for embeddings, and domain-specific guidelines to prevent representational harm.

## Scalability and memory bottlenecks

DR is essential for managing high-dimensional data, but many popular methods struggle with computational and memory bottlenecks at scale. These challenges arise from algorithmic complexity—especially pairwise computations, graph construction, and iterative optimization. In fields like genomics, geospatial analysis, and real-time sensor

networks, data volume and speed often exceed the capacity of standard DR approaches (*Xiao et al., 2023*).

### Quadratic complexity in kernel and graph-based methods

Methods such as kernel PCA, Isomap, and diffusion maps require computation of pairwise distance or similarity matrices, resulting in $O(n^2)$ or worse time and memory complexity. As datasets scale to millions of instances, these approaches become impractical. In single-cell transcriptomics, for instance, computing and storing all cell-cell similarity values can exhaust system memory before embedding even begins (*Fang et al., 2025*). Likewise, remote sensing and high-resolution imagery tasks often generate similarity matrices too large for in-memory processing without distributed systems or approximation techniques (*Wang et al., 2021a*).

### Iterative optimization in nonlinear embeddings

Nonlinear DR algorithms such as t-SNE and UMAP involve iterative optimization of complex loss functions to preserve local and global structures. These techniques become increasingly resource-intensive as dataset size increases. Low-perplexity settings in t-SNE require finer-grained neighborhood estimation, while UMAP's fuzzy simplicial set construction and stochastic gradient descent iterations scale poorly for large $n$ (*Linderman & Steinerberger, 2022*; *Taylor & Merényi, 2022*). This hampers real-time or interactive applications such as fraud detection, streaming IoT analytics, or online recommendation systems (*Ali, Borgo & Jones, 2021*).

### Memory and training overhead

Autoencoders and VAEs further compound scalability challenges. These models demand substantial computational resources, including GPU acceleration, high batch memory, and prolonged training cycles. In high-dimensional domains such as image recognition or text mining, feature vectors often exceed tens of thousands of elements—necessitating aggressive regularization, dimensional bottlenecks, or downsampling to prevent out-of-memory errors. These constraints make deep DR models infeasible in resource-constrained settings or edge devices (*Asperti & Trentin, 2020*).

### Real-world constraints and operational impact

The computational limitations of DR techniques directly impact their usability in time-sensitive and resource-limited applications. In clinical diagnostics, for example, preprocessing delays from t-SNE or autoencoder-based pipelines may hinder timely treatment decisions. In real-time urban analytics or manufacturing quality control, the inability to deploy DR models at the edge limits situational awareness and decision speed—undermining the core utility of DR.

### Summary

Scalability and memory inefficiencies remain central obstacles to the widespread adoption of DR in large-scale, real-world systems. While approximate, incremental, and distributed DR methods offer promising alternatives, their implementation and adoption remain limited in practice. To ensure DR fulfills its potential in high-volume, high-velocity

settings, future research must prioritize algorithmic scalability, low-latency inference, and hardware-efficient model architectures.

## SOLUTIONS

### Principled approaches to determining the optimal dimensionality

Determining the optimal number of dimensions ($k$) in a DR pipeline is essential for producing compact yet information-rich representations that align with the analytical task at hand. While heuristics—such as retaining components explaining 90–95% of total variance—are computationally efficient and widely adopted, they often fail to capture the trade-off between meaningful signal retention and noise suppression. A range of principled, data-driven strategies has emerged to address this challenge by tailoring DS to both the intrinsic structure of the dataset and its downstream use case (*Greenacre et al., 2022*).

#### Variance-based methods

Traditional variance-based approaches, typically applied in PCA, retain components based on cumulative explained variance. However, they rely on the assumption that variance magnitude correlates with information utility—an assumption that breaks down in domains like genomics and NLP, where low-variance features may encode critical structure. Parallel analysis refines this approach by comparing observed eigenvalues ($\lambda_i$) to those from random or permuted data ($\lambda_{\text{random}}$), retaining only components with eigenvalues exceeding this empirical null distribution (*Abdi & Williams, 2010*). This guards against overfitting due to sampling variability and improves statistical robustness.

#### Intrinsic dimensionality estimation

ID estimation seeks to quantify the minimum number of dimensions needed to represent the underlying data manifold faithfully. Techniques include correlation dimension, MLE, and geometric estimators like DANCo and TwoNN, which infer local dimensionality from distance or angle-based statistics (*Meilă & Zhang, 2024*). These approaches are well-suited to nonlinear or fractal-like datasets. However, ID estimates can be unstable in high-noise or sparsely sampled data regimes. Ensemble ID estimation—combining multiple estimators—offers improved reliability but increases computational burden. Moreover, estimated ID can guide hyperparameter tuning in DR methods, such as perplexity in t-SNE or neighborhood size in UMAP (*Van der Maaten & Hinton, 2008*).

#### Evaluation-aware and task-aligned selection

Optimal dimensionality is frequently task-dependent. For supervised learning tasks, selecting $k$ to maximize cross-validated performance metrics such as accuracy, area under the curve (AUC), or $F_1$-score is a common strategy (*Guyon & Elisseeff, 2003*). In unsupervised contexts, clustering quality metrics (*e.g.*, silhouette score, Davies–Bouldin index) and embedding integrity measures (*e.g.*, trustworthiness, continuity) can be used to evaluate candidate dimensionalities (*Li et al., 2017*). For instance, sentiment classification might succeed with $k \approx 2$, while topic modeling or semantic retrieval could require $k \geq 10$. Supervised DR approaches, including Lasso, Elastic Net, or supervised autoencoders,

further improve task alignment by filtering dimensions directly linked to outcome-relevant signals (*Bian et al., 2022*; *Zhu et al., 2013*).

### Automated hyperparameter optimization

Automated tuning frameworks—such as grid search, random search, and Bayesian optimization—can optimize $k$ in tandem with DR-specific parameters like perplexity (t-SNE), $k$-neighbors (UMAP), or regularization strength (LDA). As shown in Fig. 2, variation in perplexity and learning rate dramatically alters the structure and interpretability of t-SNE projections, underscoring the importance of proper tuning. Bayesian optimization, while more sample-efficient, can become computationally expensive on large datasets ($O(n \log n)$ per iteration). Parallel execution and GPU-accelerated evaluation are increasingly necessary for scaling these approaches to production environments (*Xiao et al., 2023*; *Van Der Maaten, 2014*).

### Ensemble and multi-scale dimensionality reduction

Ensemble DR methods aggregate insights across multiple embeddings—either with varied $k$ or distinct algorithms—to enhance stability and generalizability. Procrustes alignment can align and combine low-dimensional representations into a consensus structure (*Wang et al., 2021a*). Multi-scale DR is particularly useful in hierarchical datasets (*e.g.*, taxonomies or medical ontologies), where preserving both fine-grained and global relationships is essential. These approaches mitigate the risk of choosing suboptimal $k$ values and offer robustness when a single embedding fails to capture all relevant structures (*Healy & McInnes, 2024*).

### Summary

Optimal DR is complex and context-dependent, requiring more than simple heuristics. Variance-based thresholds and parallel analysis offer statistical rigor for linear DR, while ID estimation captures nonlinear structure. Supervised and evaluation-aware methods improve alignment with tasks, and ensemble approaches boost robustness. Integrating these into scalable workflows ensures DR outputs are both efficient and meaningful across varied data science applications.

## Strategies for robustly handling noise, outliers, and missing data

Noise, outliers, and missing data introduce significant risks into DR, often leading to distorted embeddings, spurious clustering, and compromised downstream inferences. Unlike generic preprocessing, robust DR pipelines require integrated strategies that directly address these imperfections during embedding. Addressing them systematically is essential to preserve the integrity of the underlying data manifold and ensure trustworthy analysis (*Gao et al., 2021*; *Wang et al., 2022*).

### Context-aware preprocessing

Initial preprocessing steps such as log transformation, variance-stabilizing normalization, or low-pass filtering can attenuate high-frequency noise. Winsorization and percentile clipping are effective for minimizing the influence of outliers, especially in skewed sensor or financial datasets. However, overzealous filtering may suppress valuable

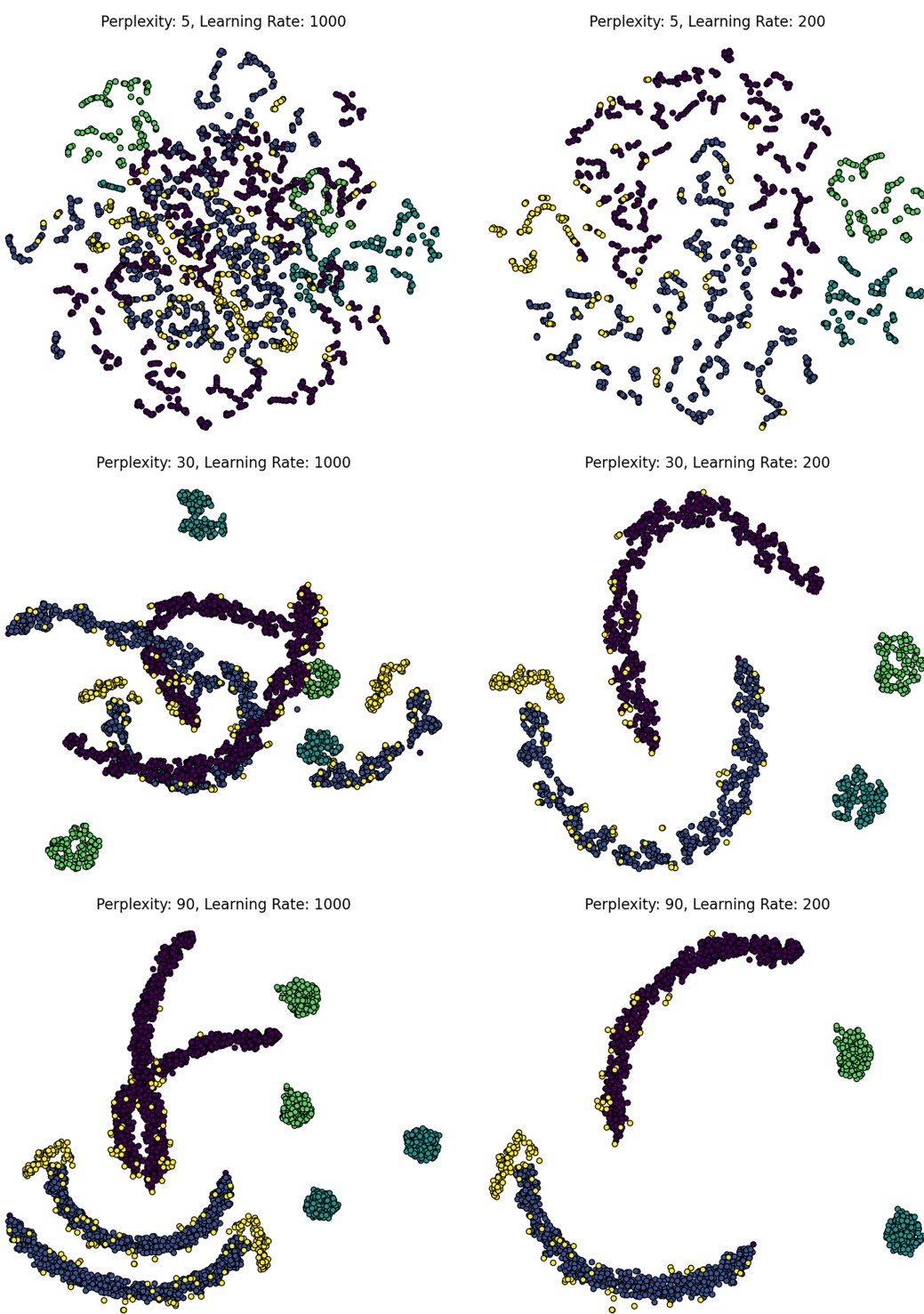

**Figure 2 Effect of t-SNE hyperparameters on MNIST.** t-SNE projections of MNIST under varying perplexity (rows: 5, 30, 90) and learning rates (columns: high *vs*. low). Low perplexity emphasizes local detail; higher values enhance global structure. High learning rates, especially with low perplexity, cause distortion and fragmented clusters. The figure highlights t-SNE's sensitivity to hyperparameter tuning and the balance between local and global structure.

signal—particularly in domains, where anomalies carry crucial information (*Ayesha, Hanif & Talib, 2020*; *Bian et al., 2022*). Even modest noise can distort decision boundaries, highlighting the need for task-aware noise mitigation. Preprocessing should balance signal preservation with effective noise reduction to maintain embedding integrity.

### Robust linear DR techniques

Robust PCA is a cornerstone method for separating structure from noise. It decomposes the data matrix into a low-rank matrix and a sparse anomaly matrix *via* the convex optimization. Weighted PCA further enhances robustness by down-weighting unreliable or high-variance observations (*Guerra-Urzola et al., 2021*). These variants preserve linear interpretability while offering improved resistance to noise and outliers.

### Improving embedding robustness

Nonlinear DR methods are highly sensitive to noise during neighborhood graph construction. Preprocessing with PCA or denoising autoencoders can suppress irrelevant variance, improving neighborhood accuracy. Regularization techniques—such as pruning weak edges, weighting edges by confidence, or penalizing local geometric variance—help stabilize graph topology and preserve manifold structure (*McInnes, Healy & Melville, 2018*; *Wang et al., 2021a*). As shown in Fig. 3, even small changes in parameter can dramatically reshape latent space, highlighting the need for careful tuning. These strategies are especially valuable where local noise can mask global patterns.

### Strategies for imputation

Missing data, if poorly addressed, can introduce bias into low-dimensional embeddings. While simple imputation is fast, it often distorts variance and degrades data quality. Soft-Impute, a matrix factorization method, offers low-rank approximations and is widely used in genomics and recommender systems (*Meyer, Pozo & Zola, 2021*). Multiple imputation approaches model uncertainty across different missingness mechanisms: MCAR, MAR, and MNAR. For high-dimensional, nonlinear data, autoencoder-based models and GANs like GAIN better capture variable dependencies and yield more accurate reconstructions (*Zhang, Chen & Hong, 2021*; *Borisov et al., 2022*). Figure 4 compares how these methods impact latent space quality.

### Domain-informed quality control

In high-stakes domains like medicine, finance, and industrial monitoring, domain-specific thresholds play a critical role. Clinical datasets may impose physiological plausibility constraints (*e.g.*, systolic blood pressure between 80–200 mmHg), while manufacturing systems define acceptable ranges for sensor variables. Incorporating these domain checks into the preprocessing pipeline helps distinguish between true anomalies and sensor noise or data entry errors, improving the validity of downstream DR outputs (*Ayesha, Hanif & Talib, 2020*).

### Summary

Reliable DR in real-world settings demands robust strategies for handling noise, outliers, and missing data. Effective DR workflows should integrate task-aware preprocessing,

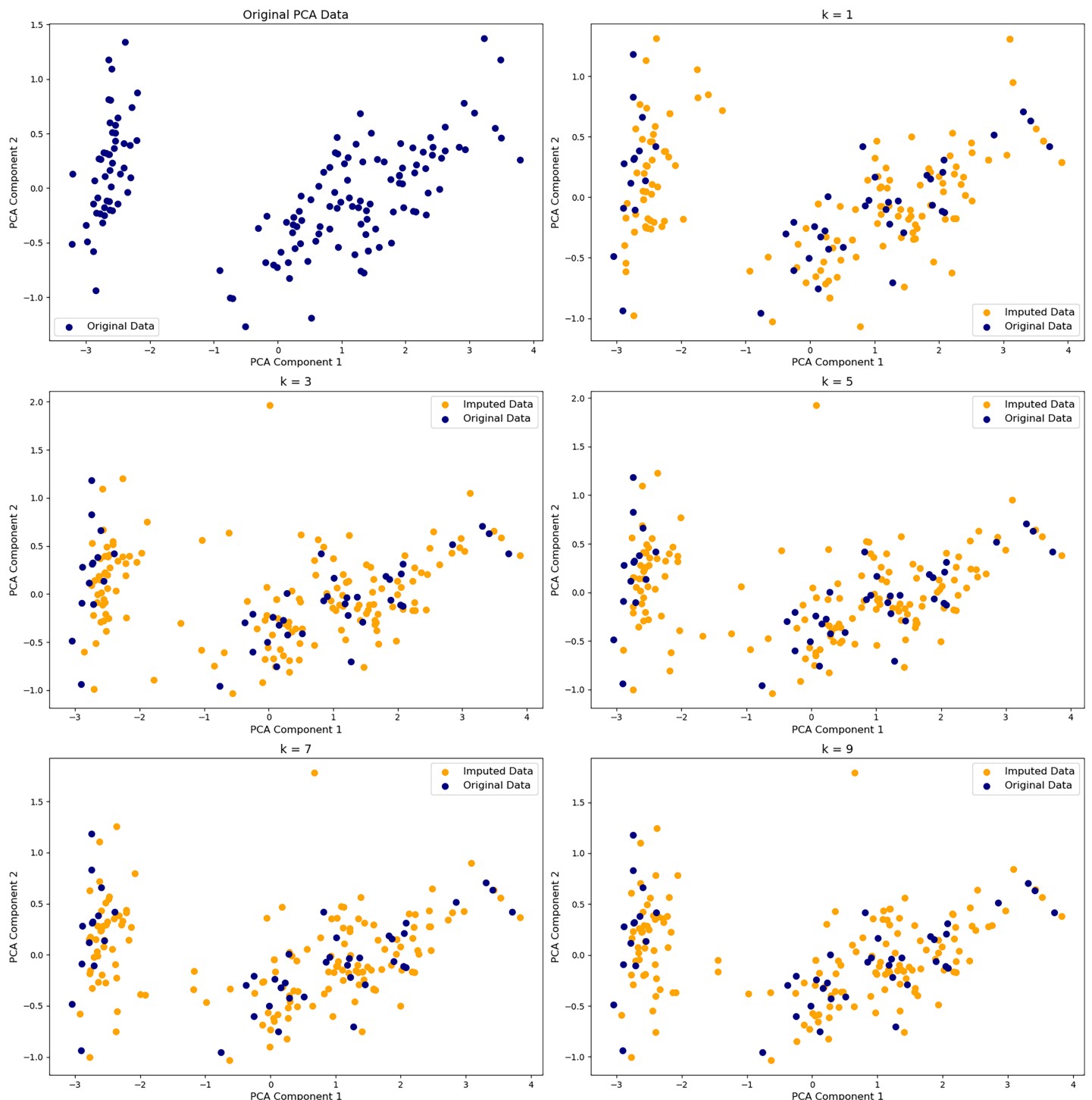

**Figure 3** **Effect of K in k-nearest neighbors (KNN) imputation on PCA projections.** PCA embeddings of the Iris dataset with 40% missing values imputed using KNN with k = 1, 3, 5, 7, 9. The top-left shows the original data. Navy points are original values; orange are imputed. Smaller k yields noisier, less stable imputations; larger k better preserves the original structure.

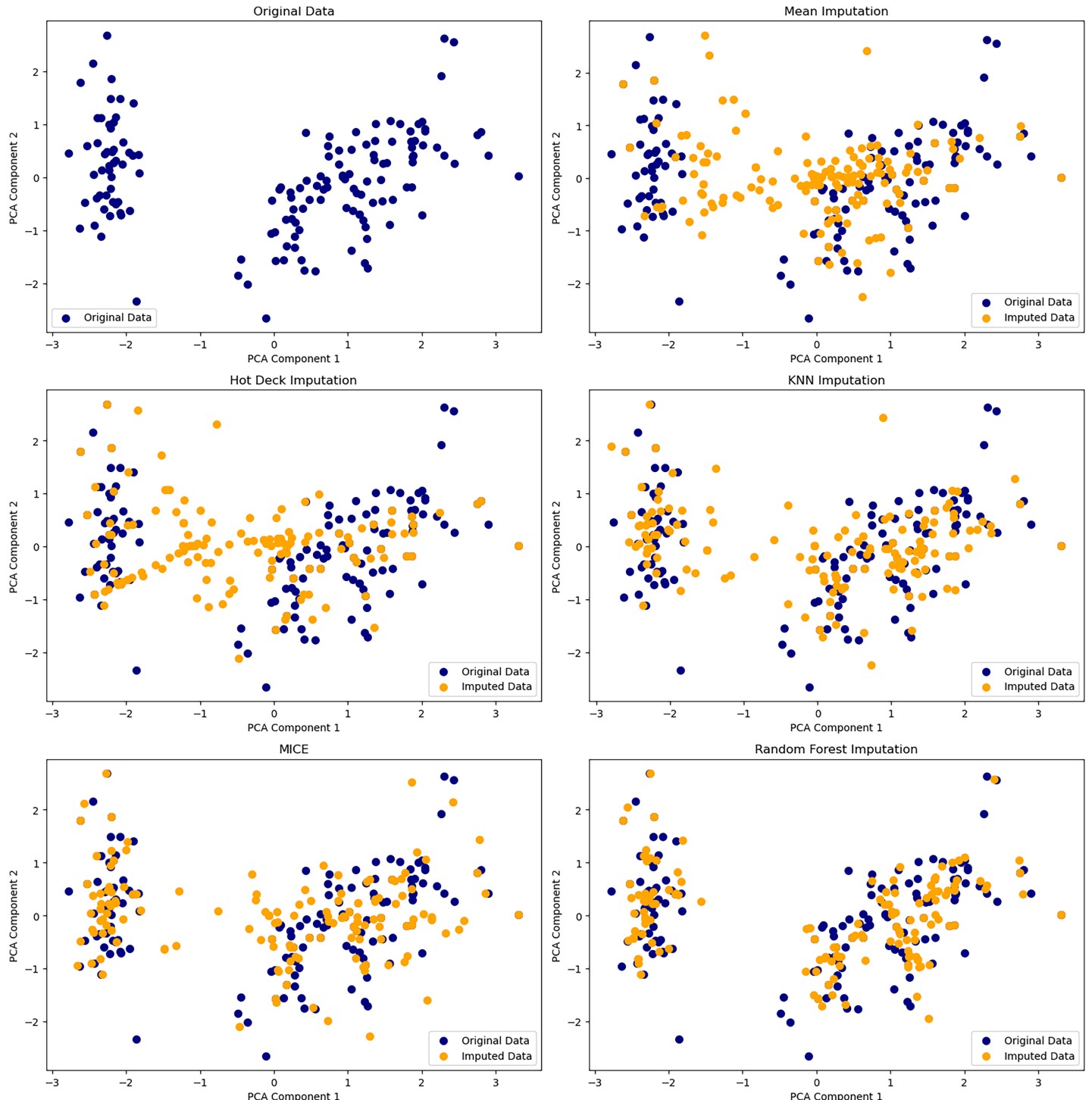

**Figure 4 Comparison of six dimensionality reduction methods—PCA, t-SNE, UMAP, Isomap, LLE, and MDS—on the Iris dataset, projecting digits into 2D.** Nonlinear methods (t-SNE, UMAP) best separate classes by preserving local structure. Isomap captures some global structure but distorts intra-class details. Linear methods (PCA, MDS) retain variance but poorly separate classes. LLE fails to unfold the manifold, compressing clusters. The comparison underscores the trade-offs between preserving structure and class separability in high-dimensional data visualization.

anomaly-resilient DR algorithms, advanced imputation methods, and domain-guided quality checks. These components collectively ensure that low-dimensional representations reflect authentic structure and support reproducible, interpretable, and context-sensitive analytics.

## Balancing interpretability with accuracy in dimensionality reduction

Modern DR techniques increasingly favor capturing complex nonlinear patterns in high-dimensional data, often at the cost of interpretability (see Fig. 5). As DR becomes integral to high-stakes applications—such as clinical decision support, fraud detection, and policy modeling—the trade-off between accuracy and transparency presents both practical and ethical challenges. Balancing these demands is essential to ensure analytically robust embeddings that also comply with regulatory standards.

### Hybrid approaches for interpretability

Combining linear and nonlinear DR techniques can yield embeddings that retain the structure of complex manifolds while maintaining a degree of interpretability. A widely used pipeline applies PCA for initial denoising and variance compression, followed by UMAP or t-SNE to extract nonlinear topological relationships (*Becht et al., 2019*). This approach benefits domains like genomics and scRNA-seq, where PCA components can be directly mapped to gene expression variance, while nonlinear methods reveal subtle phenotypic patterns (*Zhang & Lei, 2011*).

### Interpretability techniques

*Post-hoc* methods provide retrospective insight into black-box DR models. Techniques like SHAP can be adapted to estimate feature contributions to latent coordinates (*Antwarg et al., 2021*), while gradient-based saliency maps highlight influential inputs in autoencoders—useful in areas like text mining (*Borisov et al., 2022*). However, these approaches can be fragile and may not reflect true causal structure unless the DR model is well-constrained. Surrogate and perturbation-based explanations must be used cautiously to avoid over-interpretation or misleading rationales.

### Sparse and regularized embedding models

Constraining latent representations encourages interpretability. As shown in Fig. 6, increasing levels of missingness progressively distort PCA embeddings, underscoring how even modest data incompleteness can compromise interpretability in unconstrained models. Sparse AE enforce $L_1$ penalties to produce compact embeddings where each latent dimension is activated by a limited subset of inputs—particularly effective in domains such as climate modeling and metabolic profiling (*Li et al., 2023b*). $\beta$-VAEs ($\beta$ enhance interpretability by encouraging disentanglement of latent factors) through a weighted Kullback–Leibler (KL)-divergence penalty (*Kingma & Welling, 2019*). Orthogonality and independence constraints, inspired by ICA, minimize redundancy and produce latent dimensions that correspond to distinct, semantically meaningful components (*Hastie et al., 2009*).

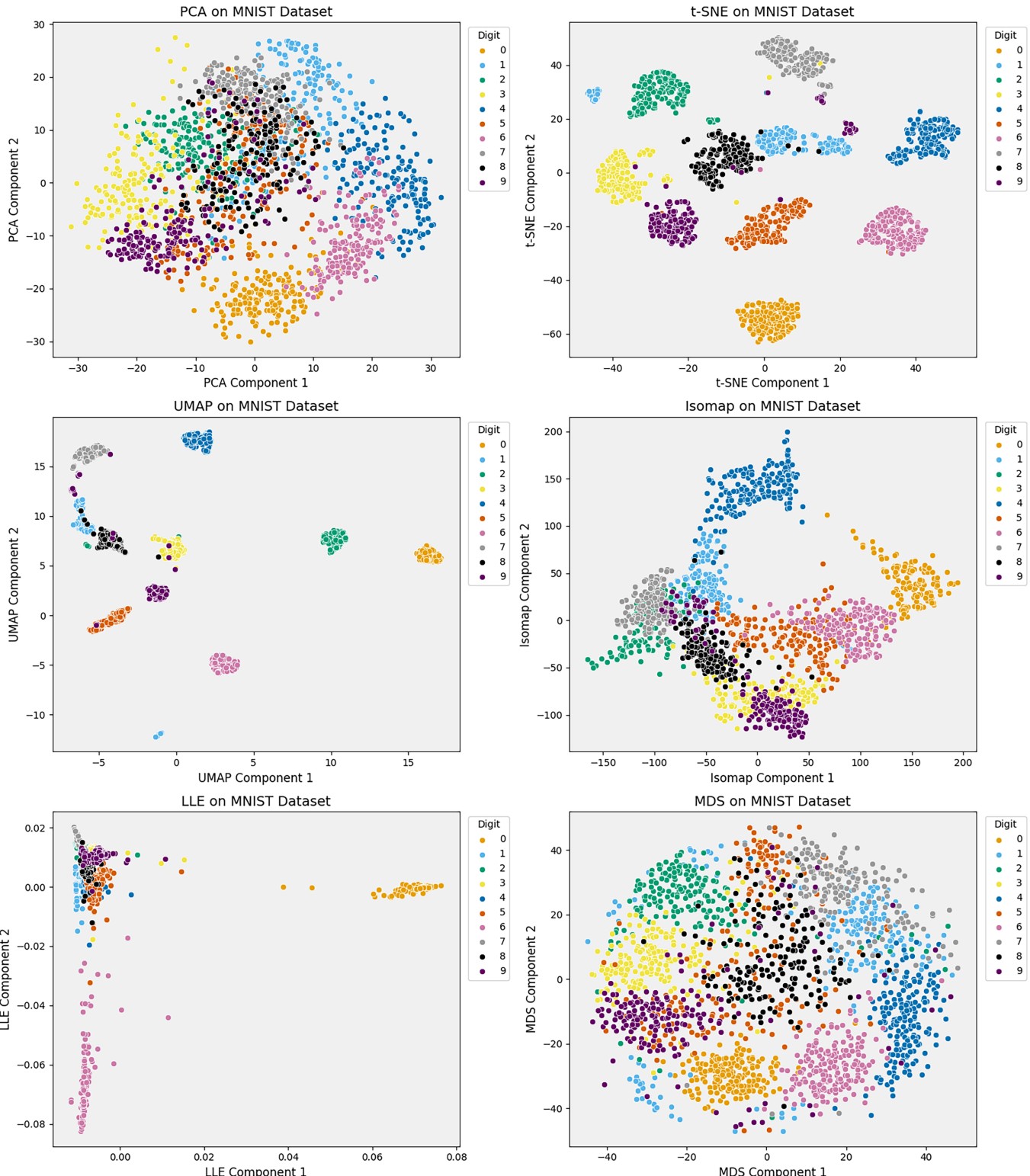

**Figure 5 Comparison of six dimensionality reduction techniques—on MNIST digit embeddings projected to 2D.** t-SNE and UMAP produce well-separated clusters, preserving local structure and class distinctions. Isomap retains some global layout but distorts intra-class details. PCA and MDS preserve variance yet blur class boundaries. LLE compresses the manifold, obscuring meaningful structure. The results underscore each method's trade-offs in visualizing high-dimensional data.  

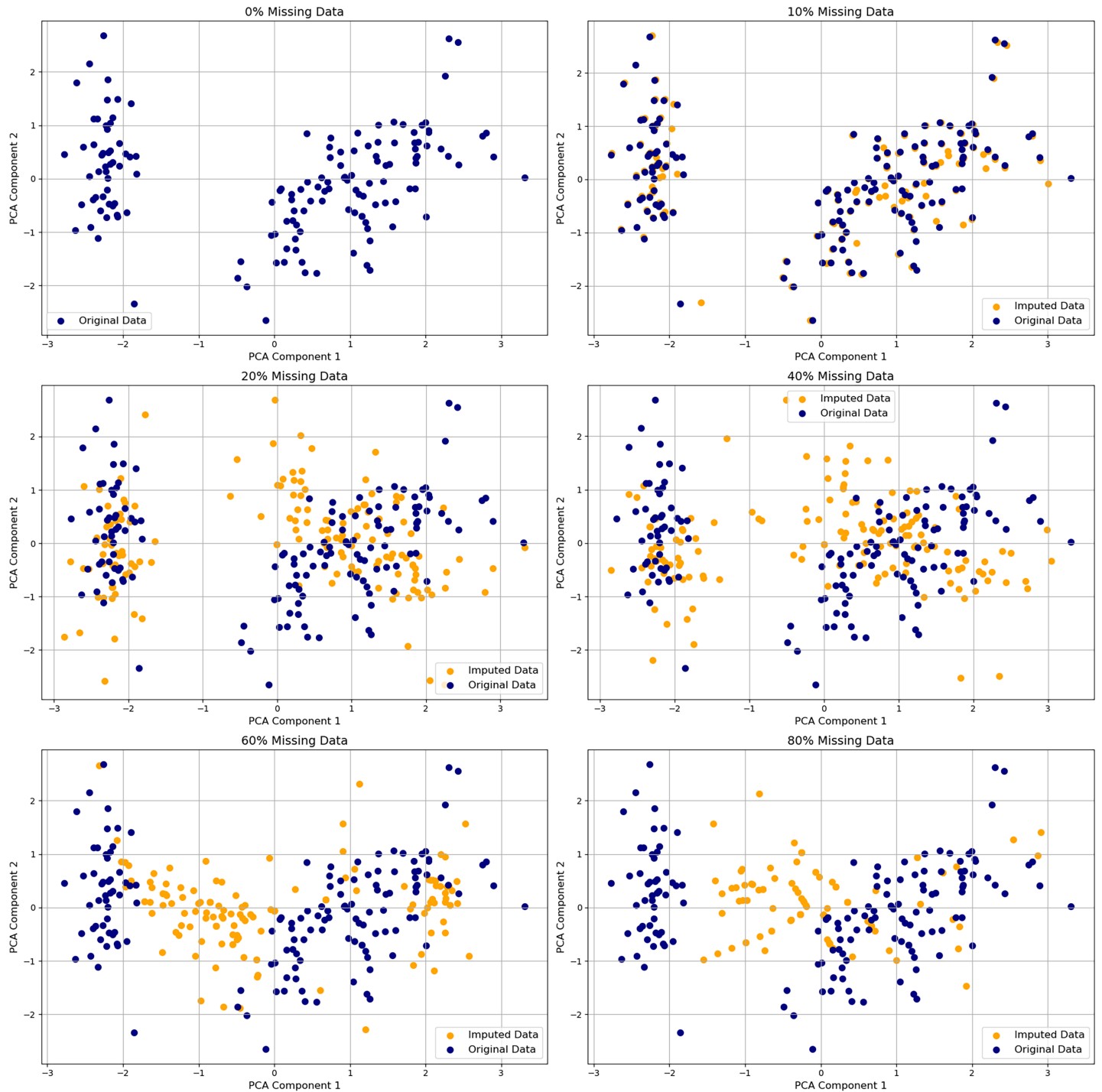

**Figure 6 Visualization of PCA on original and imputed datasets with varying percentages of missing values.** Comparison of PCA embeddings of the original Iris dataset and versions with 10–80% missing values imputed. Original data points (blue circles) and imputed values (orange crosses) show increasing divergence as missingness rises, especially beyond 40%. The plot underscores how imputation degrades geometric fidelity and warns of inflated variance or cluster drift in downstream analysis.

### Domain-informed constraints

Integrating prior knowledge through domain-informed regularization or architectural design enhances semantic interpretability. Grouping features by taxonomies—like biological pathways, financial metrics, or sensor module—helps embeddings reflect real-world structures. Applying plausibility constraints, such as energy conservation or physiological limits, keeps representations grounded and prevents overfitting to noise or spurious patterns (*Hastie et al., 2009*). These approaches strengthen the trustworthiness of DR pipelines, especially in regulated domains.

### Regulatory and operational implications

Regulatory mandates increasingly demand transparency in algorithmic decisions, including those involving DR. In healthcare, embeddings must link back to clinical features; in finance, they must justify credit risk. Requirements like the GDPR's right to explanation and IEEE standards make interpretability a compliance issue. DR pipelines using hybrid models, sparse constraints, and *post-hoc* tools—built with frameworks like scikitlearn or Keras.

### Summary

Balancing interpretability and accuracy in DR is no longer a theoretical concern—it is a practical and regulatory imperative. Hybrid pipelines, *post-hoc* explanations, sparsity-inducing models, and domain-informed constraints offer a roadmap for embedding design that is both expressive and explainable. These strategies ensure that DR outputs are not only powerful analytical tools, but also transparent, reproducible, and ethically deployable in high-stakes environments.

## Ensuring stability and reproducibility in dimensionality reduction

Low-dimensional embeddings must exhibit stability across repeated executions to support reliable interpretation, consistent decision-making, and robust scientific inference. Yet many DR techniques—particularly those involving stochastic processes yield variable results even under fixed hyperparameters, jeopardizing both replicability and reproducibility (*Kobak & Linderman, 2021*). This poses substantial risks in domains where reproducible evidence and regulatory compliance are paramount.

### Deterministic initialization and parametric models

Stability can be improved by fixing random seeds and using deterministic initialization strategies—such as initializing with PCA coordinates—to reduce run-to-run variance. Parametric DR models, including parametric t-SNE and neural network-based mappings, learn explicit transformation functions from high- to low-dimensional space, producing consistent embeddings for both existing and novel data (*Sainburg, McInnes & Gentner, 2021*). However, these methods must be deployed with careful control over environmental variables, including seed management, batch ordering, and hardware determinism.

### Ensemble methods for embedding robustness

Ensemble techniques enhance robustness by averaging or aligning results from multiple DR runs. Repeated stochastic embeddings under varying seeds can be combined using

Procrustes analysis to form consensus representations that smooth over local variability. Multi-method ensembles—integrating PCA, UMAP, Isomap, and others—offer richer representations of manifold structure, particularly valuable in exploratory tasks such as patient clustering or cross-population genomic comparisons (*Serna-Serna et al., 2023*). While computationally expensive, ensembles mitigate stochastic artifacts and improve consistency across analyses.

### Post-Hoc alignment and geometric normalization

*Post-hoc* alignment techniques address superficial differences in embeddings—such as rotation, reflection, and scaling—without retraining. Procrustes transformation aligns embeddings from different runs to a shared reference space, enabling valid comparisons. Canonical correlation analysis (CCA) further identifies common latent directions across embeddings, aiding in cross-validation and interpretability assessments. Although these techniques do not resolve deeper topological inconsistencies (*e.g.*, cluster fragmentation), they stabilize visualization and analytical coherence across stochastic outputs (*Kobak & Linderman, 2021*).

### Hyperparameter robustness

Embedding stability is highly sensitive to hyperparameters such as perplexity, neighborhood size, and learning rate. Systematic sensitivity analysis—*via* grid search, random sampling, or Bayesian optimization—can identify parameter regions that yield stable and high-quality embeddings. Stability can be quantified using metrics such as trustworthiness, continuity, neighborhood preservation, or cluster consistency across seeds (*Taylor & Merényi, 2022*). Integrating these evaluations into DR pipelines facilitates robust parameter selection and improves analytical reliability.

### Workflow versioning and environment control

Ensuring reproducibility goes beyond algorithmic solutions—it requires end-to-end workflow control. This involves version-locking libraries, logging preprocessing steps, fixing random seeds, and using infrastructure for traceability. Tools like pipeline automation, data versioning, and environment encapsulation support full-stack reproducibility. These practices are vital in regulated settings and collaborative research, where auditability and integrity are essential. Table 3 provides a practical checklist to guide reproducible DR workflows.

### Summary

Embedding stability and reproducibility are essential for responsible DR use. Techniques like deterministic initialization, parametric modeling, ensemble averaging, *post-hoc* alignment, hyperparameter tuning, and workflow versioning collectively reduce variability. While not foolproof, these strategies significantly improve the consistency and credibility of DR analyses.

## Mitigating overfitting and improving generalization

Despite being unsupervised, DR is prone to overfitting—especially in HDLSS settings marked by sparsity, noise, and heterogeneity. Such overfitting leads to embeddings that

**Table 3 Reproducibility checklist for dimensionality reduction workflows.**

| Category | Checklist Item |
| --- | --- |
| Data handling | Specify dataset origin, method of collection, all preprocessing steps (*e.g.*, scaling, filtering, normalization), and logic for train/validation/test split. |
| DR configuration | Document DR algorithm name and version, hyperparameters (*e.g.*, perplexity, neighbors, target $k$), and input feature dimensionality before reduction. |
| Stability controls | Fix random seeds for stochastic DR, report embedding variance across runs, and include quantitative stability scores (*e.g.*, trustworthiness, continuity). |
| Visualization fidelity | Ensure visualizations have labeled axes, consistent colormaps/legends, same scale/aspect ratio, and are reproducible from saved configurations. |
| Interpretability & downstream use | Save final low-dimensional embeddings, document feature attribution or diagnostics, & perform embedding inversion/influence analysis if applicable. |
| Fairness and privacy | Check for bias leakage, apply mitigation strategies (*e.g.*, adversarial debiasing), and implement differential privacy where needed. |
| Computational environment | Record compute hardware, software/library versions, and share reproducible environment files (*e.g.*, Dockerfile, 'requirements.txt'). |

capture dataset-specific artifacts rather than underlying structure, resulting in misleading clusters, poor generalization, and reduced trust in applications like personalized medicine, real-time IoT analytics, and genomics (*Bian et al., 2022*). To mitigate this, robust DR pipelines must integrate regularization, cross-validation, noise-resilient neighborhood graphs, domain-informed inductive biases, and mechanisms for transfer learning.

### Regularization and sparse encoding

Regularization is critical to preventing overfitting in high-capacity models such as autoencoders. $L_1$ regularization promotes sparse latent representations by enforcing feature selectivity, while $L_2$ regularization penalizes large weights, thereby smoothing learned transformations. Dropout further enhances generalization by randomly omitting units during training, encouraging the model to develop redundant, distributed representations (*Li et al., 2023b*). These constraints limit overfitting to noise and guide DR models toward stable, interpretable embeddings that retain semantic relevance across datasets.

### Cross-validation and stability assessment

Adapted cross-validation strategies allow indirect evaluation of embedding robustness. Reconstruction error (for PCA, autoencoders), continuity (manifold), and clustering consistency (*e.g.*, silhouette score variance across splits) can reveal overfitting tendencies. In unsupervised settings, techniques such as bootstrap neighborhood overlap, subsample stability, and manifold distortion metrics provide proxies for generalization performance (*Marukatat, 2023*). These assessments inform hyperparameter tuning and offer critical insights into the structural fidelity of embeddings.

### Robust neighborhood graph construction

Manifold-based DR algorithms rely on neighborhood graphs, which are often distorted by noise or irregular sampling densities. Enhanced robustness can be achieved through graph

trimming (removing low-confidence edges), adaptive kernels (adjusting bandwidth based on local density), and sparsification (retaining only stable connections) (*Kobak & Linderman, 2021*; *Healy & McInnes, 2024*). These refinements reduce susceptibility to noise-induced artifacts and promote more faithful recovery of the underlying data geometry.

### Embedding domain-specific inductive biases

Incorporating domain knowledge into DR models improves generalization and interpretability. Grouping features based on known ontologies—such as gene pathways, device types, or regulatory hierarchies—introduces soft structural priors. Explicit constraints, including non-negativity, conservation laws, or unit normalization, limit the embedding space to feasible configurations. These inductive biases ground embeddings in real-world semantics and ensure alignment with domain expectations, thereby mitigating overfitting to irrelevant variance.

### Incremental learning for generalization

Generalization across new or evolving datasets requires DR methods to support transferability and adaptability. Parametric techniques—such as AE and parametric t-SNE—learn explicit embedding functions, enabling out-of-sample projection without retraining (*Sainburg, McInnes & Gentner, 2021*). UMAP supports transformation of unseen data using precomputed graph structure. In dynamic environments, incremental DR models can update embeddings in response to data streams, avoiding full recomputation and maintaining temporal consistency. These properties are essential in real-time or longitudinal settings such as fraud detection and clinical monitoring (*Ali, Borgo & Jones, 2021*).

### Summary

Mitigating overfitting and improving generalization in DR requires a multifaceted strategy: regularization and sparsity reduce complexity; unsupervised validation checks embedding robustness; improved graph construction enhances noise resistance; domain biases ensure semantic coherence; and transfer learning broadens applicability. Together, these methods produce stable, interpretable, and deployment-ready DR outputs.

## Fairness and privacy-aware dimensionality reduction

As DR methods see growing use in sensitive domains, concerns about fairness and privacy have become critical. Although DR may seem privacy-preserving due to data abstraction, evidence shows it can retain, amplify, or even expose structural biases and enable adversarial reconstruction of sensitive information. Addressing these risks demands explicit integration of fairness-aware objectives and privacy-preserving mechanisms into DR pipelines to ensure ethical and compliant downstream tasks (*Prayitno et al., 2021*).

### Fairness-conscious preprocessing

Bias mitigation in DR starts with preprocessing strategies aimed at minimizing the impact of sensitive attributes. Techniques like sample reweighting, adversarial debiasing, and orthogonal projection help decorrelate protected attributes from other features. Fair

autoencoders, for example, use adversarial branches to predict sensitive traits, pushing the encoder to obscure subgroup distinctions. This has shown success on datasets like COMPAS, reducing racial bias and improving statistical parity. Orthogonal projection methods similarly remove linear ties to socioeconomic status in clinical data, promoting fairer and more generalizable latent representations.

### Bias auditing and fairness metrics

Systematic auditing of embeddings is essential for uncovering biases encoded by DR methods. Metrics such as demographic parity difference, equalized odds, statistical parity difference and entropy-based subgroup balance provide actionable diagnostics. More nuanced metrics include KL divergence between subgroup distributions and silhouette scores stratified by subgroup, both of which can detect subtle, latent-space biases. Implementing these audits using standardized libraries like fairlearn, aif360 enables consistent and transparent fairness evaluation (*Mehrabi et al., 2021*).

### Differential privacy and noise injection

To protect individual privacy, especially in sensitive applications involving personally identifiable information, differential privacy (DP) methods have been incorporated into DR algorithms. DP-compliant variants of PCA, UMAP, and t-SNE introduce calibrated noise into computations (*e.g.*, covariance matrices, gradient updates, pairwise distances), limiting the potential for re-identification. While DP mechanisms slightly compromise embedding quality, practitioners can manage this privacy-utility trade-off *via* adjustable privacy parameters ($\epsilon$), ensuring compliance with regulatory frameworks such as HIPAA, GDPR, and the EU AI Act (*Prayitno et al., 2021*).

### Embedding inversion resistance and obfuscation

DR methods, particularly linear transformations and autoencoders, inherently risk adversarial inversion, potentially compromising sensitive inputs. Techniques to mitigate inversion risks include using contractive autoencoders, random projections, dropout noise, or applying non-invertible transformations. Practitioners should perform regular inversion risk assessments using model inversion attacks or reconstruction benchmarks to quantify the vulnerability of their embeddings and iteratively enhance their resistance to adversarial reconstruction.

### Federated and decentralized DR architectures

In scenarios where centralized data aggregation poses significant privacy risks, federated and decentralized DR methods offer viable alternatives. Techniques such as federated PCA and distributed autoencoders enable institutions to locally generate embeddings, sharing only aggregated or anonymized latent representations. When coupled with DP mechanisms and secure aggregation protocols, these methods support collaborative insights without direct raw data exposure, adhering to principles of privacy-by-design and data minimization, especially vital in multi-center clinical research and international financial systems (*Prayitno et al., 2021*).

**Table 4 Rules for robust and responsible dimensionality reduction.**

| # | Best Practice | Rationale |
| --- | --- | --- |
| 1 | Audit bias before and after DR | Evaluate whether protected attributes are encoded in the latent space using metrics such as demographic parity or KL divergence. |
| 2 | Use fairness-aware preprocessing | Apply orthogonal projection, sample reweighting, or adversarial debiasing to remove correlations with sensitive attributes. |
| 3 | Choose DR method based on task goal | Linear methods are suited for interpretability, nonlinear methods for visualization, and hybrid methods for structure preservation. |
| 4 | Avoid over interpreting plots | 2D embeddings can be misleading—complement with quantitative metrics and downstream validation. |
| 5 | Quantify information loss | Use explained variance, reconstruction error, or task-specific performance to measure trade-offs. |
| 6 | Stabilize stochastic DR methods | Fix seeds, use ensemble runs, or apply consensus strategies for reproducibility (*e.g.*, t-SNE). |
| 7 | Integrate differential privacy early | Use DP-UMAP, DP-PCA, or noise-aware architectures to protect individuals in sensitive datasets. |
| 8 | Assess vulnerability to inversion | Test for re-identification using model inversion or adversarial reconstruction techniques. |
| 9 | Adopt federated DR when needed | For distributed or privacy-sensitive datasets, use federated PCA or distributed autoencoders. |
| 10 | Document all parameters and audits | Ensure transparency by logging preprocessing, seeds, hyperparameters, and audit results for reproducibility. |

*Summary*

Fairness in DR requires more than abstraction—it demands a concrete framework that includes bias-aware preprocessing, privacy protections (*e.g.*, DP), inversion resistance, and decentralized design. Practitioners must balance fairness, accuracy, and interpretability using clear metrics and practical tools to ensure ethical, trustworthy, and compliant DR in high-stakes applications.

## FUTURE DIRECTIONS

As DR evolves, future efforts will prioritize scalability, interpretability, and ethical integrity. Emerging approaches—like transformers, neural differential equations, and quantum DR—offer promise for complex or high-speed data, though some remain early-stage. Standardized benchmarks, reproducible protocols, and fairness-aware objectives are essential to address bias and privacy concerns. Scalable solutions such as incremental UMAP, GPU-accelerated t-SNE, and federated DR will support real-time, privacy-sensitive use. Finally, human-centric tools—like interactive visualizations and attribution methods—will be key to ensuring transparency and trust.

## CONCLUSIONS

DR remains a cornerstone of modern data science, enabling interpretable, efficient, and scalable analysis across high-dimensional domains. This review has synthesized the evolution of DR methods—from classical projections to deep, hybrid, and ensemble techniques—and examined their limitations across foundational challenges. We outlined actionable solutions and condensed them into rules (Table 4) for responsible and effective DR deployment. As these methods increasingly drive decision-making in sensitive

contexts such as healthcare, finance, and policy, aligning DR with principles of fairness, privacy, and reproducibility is no longer optional—it is essential. Looking forward, innovations like transformer-based embeddings, federated DR, and quantum-accelerated pipelines will shape the next generation of interpretable and ethically grounded DR. DR must be reimagined not just as preprocessing, but as a central tool for deriving meaningful, trustworthy insights from complex data.

## ACKNOWLEDGEMENTS

The authors acknowledge the use of an AI tool, specifically ChatGPT, to support language refinement and content organization. Final decisions and all intellectual contributions remain those of the author.

### Funding

The author has received no funding for this work

### Competing Interests

The author declares that they have no competing interests.

### Author Contributions

- Aasim Ayaz Wani conceived and designed the experiments, performed the experiments, analyzed the data, performed the computation work, prepared figures and/or tables, authored or reviewed drafts of the article, and approved the final draft.

### Data Availability

The code is available in the Supplemental Files.

The Iris dataset is available at: Fisher, R. (1936). Iris [Dataset]. UCI Machine Learning Repository. https://doi.org/10.24432/C56C76.

### Supplemental Information

Supplemental information for this article can be found online at http://dx.doi.org/10.7717/peerj-cs.3025#supplemental-information.

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
