# Peer review of "Comprehensive review of dimensionality reduction algorithms: challenges, limitations, and innovative solutions"

_PeerJ Computer Science, doi:10.7717/peerj-cs.3025_

## Round 0.1 · original submission · Major Revisions

The authors Appealed the earlier decision and we believe they should be given the opportunity to respond to the current reviewers comments. Please provide a detailed revision and rebuttal and when it is received, we may choose to invite additional reviewers.

· Appeal

Appeal


· · Academic Editor

Reject

While the paper covers a wide array of techniques, it lacks empirical comparisons. For example, there is no extensive experimental section showing how these methods perform across various datasets. A side-by-side comparison of these methods' performance on real-world data would be beneficial.

The paper leans heavily on theoretical explanations, often at the expense of practical guidelines. For practitioners seeking to apply these methods, the paper could provide more concrete recommendations based on dataset characteristics (e.g., when to prefer t-SNE over PCA in practice).

Techniques like t-SNE and UMAP are known to be highly sensitive to hyperparameters (e.g., perplexity in t-SNE and the number of neighbors in UMAP), which can dramatically change the results. While this is mentioned, the paper could delve deeper into the best practices for tuning these parameters to avoid misleading visualizations.

Many real-world applications, such as in genomics or text data, involve high-dimensional datasets but with relatively few samples. The paper could benefit from a more focused discussion on techniques that perform well in these settings, such as Regularized PCA or autoencoders specifically designed for small sample sizes.

Though the paper discusses computational complexity, it does not provide enough solutions for handling large-scale datasets. Methods like t-SNE and Isomap are computationally expensive, and the paper could offer more practical strategies (e.g., approximations, distributed computing) to overcome scalability bottlenecks.

While the paper acknowledges that methods like t-SNE and UMAP focus on local data structure at the expense of global patterns, it does not offer much discussion on hybrid methods or how to choose between local and global techniques depending on the problem. It could benefit from a deeper exploration of strategies for preserving both types of structure.

Reviewer 1 ·

Basic reporting

The paper is well-structured, uses clear, and adheres to academic standards. The literature review is thorough, referencing seminal works and recent advancements in the field of dimensionality reduction. Figures are high quality, well-labeled, and effectively aid in illustrating complex concepts.

Experimental design

The research question is well-defined and relevant, addressing the theoretical aspects of dimensionality reduction techniques. The paper does a commendable job of categorizing and comparing various methods, making it a valuable resource for both practitioners and researchers.

But the paper predominantly relies on theoretical analysis without empirical validation, which could limit the practical applicability of the findings.

Validity of the findings

Good Points:
The paper effectively highlights the strengths and weaknesses of each technique, providing valuable insights for both practitioners and researchers.

The theoretical analysis is rigorous, with detailed discussions on the mathematical foundations and computational aspects of the techniques reviewed.

Improvements:
Some of the conclusions are drawn broadly based on theoretical analysis without sufficient empirical support, which may not hold in practical scenarios.

Where possible, support theoretical claims with data from real-world applications or simulations to enhance the validity of the findings.

Additional comments

I recommend acceptance with minor revisions. The author should consider incorporating more examples of real-world applications or case studies to enhance the practical relevance of the paper. Additionally, some sections, particularly those on newer techniques like UMAP and t-SNE, could be expanded to include more about their performance in real datasets compared to traditional methods.

I have some comments about the mathematical formulas:
- Equation 3 : Y=X V_k ( k should be written as index) / Explained Variance, the indices of the sum of the eigenvalues in the denominator should be fixed.
-Equation 19 (Weight Vector Update) : wi(t +1) = wi(t)+η(t)hci(t)(x2wi(t)). Indices should be fixed

·

Basic reporting

1. A review is good written but need a brief review regarding English grammar and composition.
2. Formula and equations should be checked by the authors to ensure that variable and corresponding values have been justified properly in explanation.
3. Figures seems right
4. Comparison table is missing
5. Coding and implementation details missing

Experimental design

Equations and formula need cross check before final submission

Validity of the findings

1. Suitability criteria for the selection of Dimensions and dimensionality reduction techniques is missing

Additional comments

No

Reviewer 3 ·

Basic reporting

The article titled "Advances in Dimensionality Reduction: In-depth analysis of linear and nonlinear techniques for machine learning" aims to provide a comprehensive analysis of dimensionality reduction techniques, covering both linear and nonlinear methods. This is an important topic in the field of machine learning. However, it fails to introduce any new perspectives on the subject of dimensionality reduction. Moreover, most of the content are well-known information already extensively covered in the existing literature. In my opinion, this is just review of existing literature

Experimental design

The article has experimental results which were on some data. These kind of results are already available on different libraries. This article should include experiments that compare various dimensionality reduction techniques across different real world datasets. There is no significant novel contribution that adds value to the current state of research.

Validity of the findings

The practical challenges associated with dimensionality reduction, such as computational cost, the curse of dimensionality, and issues related to interpretability and parameter tuning, should be properly discussed for different datasets. These are important aspects that need to be considered, especially when applying these techniques in real-world scenarios.

Additional comments

No comment

Reviewer 4 ·

Basic reporting

The paper provides a well-written and comprehensive overview of dimensionality reduction techniques, and the English language used is clear and professional. The introduction does a good job of establishing the importance of dimensionality reduction in machine learning and data science. The figures are generally clear and of high quality.

Experimental design

The paper provides an extensive review of both linear and nonlinear dimensionality reduction techniques.The methods are well-explained with detailed descriptions of the mathematical foundations of each technique. The structure and presentation of the methods are detailed and organized effectively. The categorization of linear and nonlinear techniques is particularly useful for readers trying to understand which approach to use in different scenarios.

Validity of the findings

The paper provides a solid and insightful analysis of each technique’s computational requirements, mathematical foundations, and limitations. The comparisons between methods are clear and based on sound reasoning. The conclusions are consistent with the research questions posed in the introduction, offering a well-rounded evaluation of the techniques discussed.

Additional comments

The paper provides a comprehensive and well-organized review of both classical and modern dimensionality reduction techniques, covering a wide spectrum from PCA to autoencoders and UMAP.
The mathematical rigor is a key strength, as it offers readers a deep understanding of the underlying principles of each technique. This makes the paper an excellent resource for both academics and practitioners.

---

## Round 0.2 · Minor Revisions

**Language Note:** The review process has identified that the English language must be improved. PeerJ can provide language editing services - please contact us at [email protected] for pricing (be sure to provide your manuscript number and title). Alternatively, you should make your own arrangements to improve the language quality and provide details in your response letter. – PeerJ Staff

Reviewer 1 ·

Basic reporting

The paper is well-written in clear and unambiguous English. The background context is comprehensive, with extensive references to relevant and recent literature across the fields of machine learning, statistics, and domain-specific applications such as bioinformatics and NLP. The review successfully provides a self-contained discussion, with all key terms and concepts clearly defined.

the structure of the article follows a logical progression, from taxonomy and methodological categorization to a deep dive into challenges and solutions . Plots, figures are well-labeled and informative, and their quality is very good, though a visual taxonomy diagram would further improve clarity and reader navigation. The manuscript could also benefit from a brief summary table mapping each DR technique to its primary strengths and weaknesses.

Regarding tables 1,2,3,4,5 , they are very dense, and the mathematical formulas requires an additional explanation of these equations and and a brief description for the different terms in teh equations.

Experimental design

As a review paper, the article does not report on novel experimental research, which is appropriate for the journal’s scope when offering a critical and integrative synthesis of a research area.
The article clearly states its research question, to evaluate dimensionality reduction techniques in light of eight persistent challenges (for example: dimensionality selection, interpretability, privacy). This is relevant addressing a clear gap in the literature by connecting technical limitations to real-world concerns like fairness and regulatory compliance.

The methodological approach consisting of categorizing DR techniques and discussing them in the context of specific challenges, is rigorous and intellectually sound. The inclusion of domain-specific examples (for example: scRNA-seq, fraud detection, EHR data) adds depth and supports generalizability and reproducibility.

Overall, given that this is review paper, I think that the review and comparison were performed to a high technical standard.

Validity of the findings

Regarding the validity of teh findings, the findings are consistent with existing literature and offer thoughtful, balanced interpretations. Though no original experiments are performed, the paper synthesizes evidence from a large body of peer-reviewed research and identifies trends, gaps, and promising future directions. The conclusions are well-stated and don't overreach beyond the scope of the review.

The manuscript would be strengthened by a deeper discussion of empirical benchmarking studies, either by including comparative results from previous work or suggesting standard evaluation frameworks. This would further validate the recommendations for practitioners.

Additional comments

I think that this second version of the paper is much better than the initial version, the changes made by authors were very clear in this improved version.

Great job!

·

Basic reporting

Make changes successfully.

Experimental design

Corrected

Validity of the findings

Good

Additional comments

Accept

Reviewer 3 ·

Basic reporting

The manuscript titled "Comprehensive Review of Dimensionality Reduction Algorithms: Challenges, Limitations, and Innovative Solutions" a thorough and well-structured review of dimensionality reduction techniques, covering both traditional and modern approaches.

- There are many grammatical mistakes. Authors should consider when submitting revised version. For example, do not use & within the sentence (See line #76, 78)

- Citation format within the text is not correct. These should be like (Authors, year) if written at the end of senetence.
-

Experimental design

Author has performed visualization of different dimensionality reduction techniques. However, the impact of these dimensionality reduction techniques on different machine learning models is missing. Furthermore, author has not compared the results of different techniques quantitatively.

Validity of the findings

The findings should be compared quantitatively.

---

## Round 0.3 · accepted · Accept

Congratulations, your paper has been accepted for publication.